

# Applicability and perspectives for DNA barcoding of soil invertebrates

Jéhan Le Cadre[1,2], Finn Luca Klemp[1], Miklós Bálint[3,4], Stefan Scheu[1,5] and Ina Schaefer[1,3,4]

[1] J. F. Blumenbach Institute of Zoology and Anthropology, University of Göttingen, Göttingen, Germany
[2] Biocenter, Ludwig-Maximilians-Universität München, Planegg-Martinsried, Germany
[3] Senckenberg Biodiversity Climate Research Center, Frankfurt Main, Germany
[4] Loewe Center for Translational Biodiversity Genomics (LOEWE-TBG), Frankfurt Main, Germany
[5] Centre of Biodiversity and Sustainable Land Use, University of Göttingen, Göttingen, Germany

Corresponding author
Jéhan Le Cadre,
jehanlecadre@gmail.com

## ABSTRACT

Belowground invertebrate communities are dominated by species-rich and very small microarthropods that require long handling times and high taxonomic expertise for species determination. Molecular based methods like metabarcoding circumvent the morphological determination process by assigning taxa bioinformatically based on sequence information. The potential to analyse diverse and cryptic communities in short time at high taxonomic resolution is promising. However, metabarcoding studies revealed that taxonomic assignment below family-level in Collembola (Hexapoda) and Oribatida (Acariformes) is difficult and often fails. These are the most abundant and species-rich soil-living microarthropods, and the application of molecular-based, automated species determination would be most beneficial in these taxa. In this study, we analysed the presence of a barcoding gap in the standard barcoding gene *cytochrome oxidase I* (COI) in Collembola and Oribatida. The barcoding gap describes a significant difference between intra- and interspecific genetic distances among taxa and is essential for bioinformatic taxa assignment. We collected COI sequences of Collembola and Oribatida from BOLD and NCBI and focused on species with a wide geographic sampling to capture the range of their intraspecific variance. Our results show that intra- and interspecific genetic distances in COI overlapped in most species, impeding accurate assignment. When a barcoding gap was present, it exceeded the standard threshold of 3% intraspecific distances and also differed between species. Automatic specimen assignments also showed that most species comprised of multiple genetic lineages that caused ambiguous taxon assignments in distance-based methods. Character-based taxonomic assignment using phylogenetic trees and monophyletic clades as criteria worked for some species of Oribatida but failed completely for Collembola. Notably, parthenogenetic species showed lower genetic variance in COI and more accurate species assignment than sexual species. The different patterns in genetic diversity among species suggest that the different degrees of genetic variance result from deep evolutionary distances. This indicates that a single genetic threshold, or a single standard gene, will probably not be sufficient for the molecular species identification of many Collembola and Oribatida taxa. Our results also show that haplotype diversity in some of the investigated taxa was not even nearly covered, but coverage was better for Collembola than for Oribatida. Additional use of secondary barcoding genes and long-read sequencing of marker genes can improve metabarcoding studies. We also recommend the construction of pan-genomes and pan-barcodes of

species lacking a barcoding gap. This will allow both to identify species boundaries, and to cover the full range of variability in the marker genes, making molecular identification also possible for species with highly diverse barcode sequences.

# INTRODUCTION

Soils are among the most diverse habitats on earth, harbouring 25% to 50% of the biodiversity on Earth (*Decaëns et al., 2006*; *Decaëns, 2010*; *Anthony, Bender & van der Heijden, 2023*). This biodiversity drives essential processes for life on Earth and provides ecosystem services that impact human wellbeing, such as the decomposition of dead organic material, recycling of nutrients and carbon storage (*Wardle et al., 2004*; *Lavelle et al., 2006*; *Bardgett & van der Putten, 2014*). Characterizing and monitoring soil biodiversity therefore is of general interest to maintain and preserve soil functions (*Orgiazzi et al., 2015*). However, this is a challenging and time-consuming task due to the enormous taxonomic diversity and cryptic lifestyles of soil-organisms. Molecular methodologies offer great advantages for soil biodiversity assessment in terms of time and cost efficiency, and taxonomic resolution (*Antil et al., 2022*; *Eisenhauer, Bonn & Guerra, 2019*).

A large fraction of soil animal biodiversity is represented by microarthropods with body-sizes between 0.1 and 2 mm. Collembola (Hexapoda) and Oribatida (Acari: Sarcoptiformes) are dominant and omnipresent microarthropod taxa, and occur in all soil-related habitats where they reach high abundances of up to 50,000–100,000 individuals per square meter (*Bardgett & van der Putten, 2014*). Traditionally, Collembola and Oribatida have been described as decomposers, microbivores and fungivores, but studies using stable isotopes showed that they actually cover several trophic levels, demonstrating trophic specialization and functional diversity within these taxa (*Schneider et al., 2004*; *Pollierer et al., 2009*; *Potapov et al., 2016*; *Maraun et al., 2023*). These microarthropods spend their entire life in the soil matrix or in the litter layer, which makes them interesting candidates as bioindicators of soil quality in monitoring programs (*Gulvik, 2007*). Collembola are typical r-strategists with fast reproduction cycles, whereas Oribatida are usually considered K-strategists with long-life spans of 1–3 years and low fecundity, but species with shorter life-cycles are also common (*Maraun & Scheu, 2000*; *Pfingstl & Schatz, 2021*). The general differences in life-history traits and trophic diversity between Collembola and Oribatida could be informative for monitoring programs. Collembola respond and recover more quickly to disturbances (*Ponge et al., 2003*; *Santorufo et al., 2012*) than Oribatida, which have long recovery times and therefore are more sensitive to environmental changes (*Zaitsev et al., 2002*; *Gulvik, 2007*; *Pfingstl & Schatz, 2021*). However, the wide range of functional and life-history traits among different species necessitates species level determination in order to better understand their interactions in the soil system or to use them as bioindicators for changes in soil functions. About 9,000 species of Collembola and 11,000

species of Oribatida are described worldwide, but this likely represents only about 20 % of the expected species (*Potapov et al., 2020*; *Behan-Pelletier & Lindo, 2023*). Local species richness of these two taxa can be very high, reaching 60–100 species in forest soils (*Rusek, 1998*; *Schatz & Behan-Pelletier, 2008*). High species richness and abundance, and small body sizes of both, Collembola and Oribatida, pose a significant challenge for biodiversity assessments. Molecular applications, such as DNA barcoding and metabarcoding, have great potential to aid specimen identification and biodiversity assessment (*Valentini, Pompanon & Taberlet, 2009*). These methods utilize a standardized DNA fragment for taxonomic assignment of specimens by matching DNA sequences of undetermined individuals to a reference database (*Hebert et al., 2003*; *Hebert & Gregory, 2005*). This enables to automatically assign any taxonomic level and even species names to undetermined individuals. It is applicable to mixed samples of pooled specimens, which significantly reduces workload and costs. Further, molecular identification tools are equally applicable to juveniles that often lack taxonomic characters (*Richard et al., 2010*; *Grzywacz et al., 2021*). Automated handling of samples, simultaneous identification of multiple individuals in a single reaction, and the scalability of molecular data to any taxonomic level offers new opportunities for analysing spatial and temporal dynamics of soil-living animals (*Arribas et al., 2021*; *Decaëns, 2021*), and thereby provide new perspectives for monitoring of soil biodiversity. The method, however, relies on two preconditions: (1) a representative reference database and (2) a marker (barcoding) gene that reliably separates species. The most common databases are BOLD ("The Barcode of Life Data System", Ratnasingham & Hebert, 2007) and NCBI (https://ncbi.nlm.nih.gov/). The standard barcoding gene for Metazoa is a 658 bp region of the mitochondrial *cytochrome oxidase I* gene (COI; *Hebert et al., 2003*; *Hubert et al., 2008*). In general, a minimum of 500 bp of COI is required, but shorter fragments can also be used for specimen identification and species discovery (*Hajibabaei et al., 2006*; *Collins & Cruickshank, 2012*).

Two types of methods have been developed for molecular species assignment. Distance-based methods, such as the barcoding gap (*Hebert et al., 2003*), the BIN system of BOLD (*Ratnasingham & Hebert, 2013*) and Neighbor Joining (*Saitou & Nei, 1987*; *Hebert et al., 2003*), which transform sequence alignments into a genetic distance matrices. These genetic distances can be calculated based on the observed differences between sequences, or by including a model of sequence evolution that accounts for mutational processes. Based on a threshold value of similarity, these distances are then used to assign sequences to known species or to identify putatively new species. Alternatively, character-based methods, such as Maximum Likelihood, GMYC (*Pons et al., 2006*) and PTP (*Zhang et al., 2013*), rely on a phylogenetic tree. These approaches use DNA sequences directly without a distance matrix and information on character evolution is not getting lost by capturing differences among sequences in a single metric. These methods infer species based on branching frequencies in a time-calibrated tree (GMYC) or by comparing the number of substitutions on branches (PTP) using all characters in an alignment. Reciprocal monophyly of taxa on a Maximum Likelihood or Bayesian inference tree is also appropriate to check if sequences can be assigned to known species. The dependence on a phylogenetic tree, however, makes them computational more demanding compared to distance-based methods. Character-based

methods also require a threshold for delineating species boundaries and a model of sequence evolution to infer phylogenetic relationships. In DNA barcoding, the K2P model is the most widely used (*Hebert et al., 2003*; *Nishimaki & Sato, 2019*), and can be viewed as a compromise between observed genetic distances that do not account for evolutionary changes and complex models which might overestimate genetic distances in closely related taxa. However, the use of K2P as a standard model has also been criticized, but identification success is hardly affected, even if K2P poorly fits as model for a dataset (*Srivathsan & Meier, 2011*; *Collins et al., 2012*).

The success of species delineation based on genetic markers depends on the presence of a threshold value, also known as the barcoding gap, which implies that genetic distances within a species are smaller than genetic variances to congeneric and other species (*Meyer & Paulay, 2005*). A global threshold of 2%–3% intraspecific sequence divergence, or 10x the mean intraspecific divergence, has been proposed to reliably separate species (*Hebert et al., 2004*). Such a universal threshold is extremely helpful for automated species assignment of genetic data in bioinformatic pipelines. This threshold seems to be valid for a range of taxa (*Hebert, Ratnasingham & de Waard, 2003*; *Hebert et al., 2004*; *Barrett & Hebert, 2005*), but its universal application has been questioned for other species (*e.g.*, *Burns et al., 2007*; *Chapple & Ritchie, 2013*; *Elias et al., 2007*; *Meier et al., 2006*; *Meyer & Paulay, 2005*; *Wiemers & Fiedler, 2007*). In particular soil-living animals show high intraspecific divergences in the COI gene that commonly exceed the standard barcoding threshold. Examples cover different families of earthworms (*King, Tibble & Symondson, 2008*; *Novo et al., 2010*; *Martinsson, Rhoden & Erseus, 2016*), Collembola (*Porco et al., 2012a*; *Porco et al., 2012b*; *von Saltzwedel, Scheu & Schaefer, 2016*; *Zhang et al., 2019*) and Oribatida (*Rosenberger et al., 2013*; *von Saltzwedel et al., 2014*). These studies question the general effectiveness of COI for specimen identification in these taxa. Moreover, asexual reproduction occurs in 7–10% of all species in several families of Collembola and 10% of all species of Oribatida (*Chahartaghi, Scheu & Ruess, 2006*; *Cianciolo & Norton, 2006*; *Chernova et al., 2010*; *Bluhm, Scheu & Maraun, 2016*), and asexual species can be dominant in temperate forests (*Maraun & Scheu, 2000*). According to theory, asexual organisms accumulate mutations over time, until they go extinct due to the accumulation of too many deleterious mutations (Muller's ratchet: *Muller, 1964*; Kondrashov's hatchet: *Kondrashov, 1988*). This suggests that present day populations of asexual species represent a range of COI haplotypes, while populations of sexual species should represent discrete clusters of similar COI haplotypes, standing for independently evolving lineages that interbreed (*Barraclough, Birky jr & Burt, 2003*). In consequence, a barcoding gap should not be present in asexual species but rather a continuum of slightly divergent individuals. Further, hybridization events are potential origins of asexual species, and if followed by mitochondrial introgression the detection of a barcoding gap is difficult (*Mutanen et al., 2016*; *Dupont et al., 2016*). Altogether, asexual reproduction could blur lines of species identification, and a hybrid species could be wrongly identified as its maternal species.

The aims of this study were to test (i) if the standard barcoding marker gene COI meets the precondition to reliably assign species in Collembola and Oribatida, and (ii) the accuracy of separating species based on a barcoding gap. Many species of these taxa have

wide distribution ranges, and European species often occur across Palaearctic or Holarctic regions. Geographic coverage of samples provided in DNA barcode reference libraries can affect species assignment (*Hebert et al., 2003*). We therefore focused on species with a dense and broad geographic sampling to cover the potential range of intra-specific haplotype variation of COI (*Phillips, Gillis & Hanner, 2022*). We also included parthenogenetic (asexual) species to test (iii) if the reproductive mode affects the barcoding gap, because parthenogenetic species likely carry a continuum of divergent haplotypes due to the accumulation of mutations and the absence of homogenizing effects of mixis. We analysed the performance of COI for species delimitation using distance- and character-based methods.

## MATERIALS & METHODS

### Taxa collection

Datasets were obtained by checking literature and public databases (BOLD, NCBI). For Oribatida, BOLD delivered 12,252 records (search term: "Sarcoptiformes") with species names, which represent 710 species; NCBI delivered 29,047 records (search term: "Oribatida COI") with a sequence length between 500 and 800 base pairs. For Collembola (search term: "Collembola"), BOLD delivered 62,681 records with species names, which represent 1,544 species, and NCBI (search term: "Collembola COI") had 51,684 sequences with a sequence length between 500 and 800 base pairs. To assess intra- and interspecific genetic variance of these species, we downloaded sequences of congeneric species that were represented in databases with a minimum of 3-5 sequences per species. Many records in NCBI do not have a geographic reference, and most sequences in BOLD are from various geographic regions, predominantly coming from North America (Centre for Biodiversity Genomics). Analysing specimens from different continents could generate confounding effects due to ancient geographic isolations. To obtain a comparable dataset for all investigated species we therefore decided to restrict our analyses to sequences published by *Rosenberger (2011)*, *Rosenberger et al. (2013)*, *von Saltzwedel et al. (2014)* and *von Saltzwedel, Scheu & Schaefer (2016)* (Table 1), which have a comparable sampling across Europe. We selected five oribatid mite species (*Rosenberger, 2011*; *Rosenberger et al., 2013*; *von Saltzwedel et al., 2014*) and two Collembola species (*von Saltzwedel, Scheu & Schaefer, 2016*) that were collected across several countries in Europe. Accession numbers of all sequences used in this study are summarized in Table S1. Three of the five oribatid mite species are parthenogenetic. For both Collembola and one Oribatida species (*Oppiella nova*), sequences of the nuclear gene 28S rDNA of the same individuals were also available in NCBI and were used to check if genetic divergences are congruent between the mitochondrial and nuclear genes. The dataset of *Oppiella nova* (Oppiidae) differs as it contains sequences from different habitats collected only in Germany. Further, only a single congeneric sequence was available for this genus (*O. subpectinata*), but several sequences from species of other genera in the family Oppiidae. We decided to include this species into our analysis, but to check if a barcoding gap is present at genus level. Oppiidae species are very small, their body size typically ranges from 130 to 300 μm (except *Oppia nitens*, with a body size of >400 μm), which

makes species determination very laborious and explains why this family commonly is not resolved to lower taxonomic levels in community studies. Confirmation of a barcoding gap and accurate species delimitation at genus level for this family would be helpful for future DNA-based biodiversity assessments because Oppiidae is a species rich and very common family across many habitats, reaching high abundances and even being the dominant taxon in many oribatid mite communities (*Zaitsev et al., 2002*; *Bluhm, Scheu & Maraun, 2016*).

For Collembola, we selected geographically comparable datasets for two sexual species, *Folsomia quadrioculata* and *Ceratophysella denticulata*. We did not include the parthenogenetic Collembola *Parisotoma notabilis* in our analyses, which is also represented with a Europe-wide sampling, because multiple genetic lineages (cryptic species) have already been reported for this species (*Porco et al., 2012a*; *von Saltzwedel, Scheu & Schaefer, 2017*). *Isotomiella minor*, another parthenogenetic Collembola species, was excluded because congeneric sequences in the reference databases were inadequate for this study (only two sequences of *Isotomiella* sp. and one sequence of *I. paraminor*). The Collembola of the genus *Lepidocyrtus* were omitted, because this genus has been reported to be a species complex (*Cicconardi et al., 2010*) with uncertain status of the species *L. cyaneus,* which appears to be polyphyletic within *L. lanuginosus* (*Zhang et al., 2018*; *Zhang et al., 2019*).

## Species delimitation

First, we downloaded congeneric taxa from BOLD and NCBI. Second, species assignment and barcoding gap analyses were performed with two global datasets, including all Collembola and Oribatida species, respectively. Third, for a more detailed analysis, the global datasets were separated into local datasets, each comprising all sequences of a genus (family in Oppiidae).

All sequences of a genus, and all sequences of Oppiidae were aligned separately in AliView v1.28 (*Larsson, 2014*) using default settings and trimmed to the approximately shortest sequence. For the global dataset, all alignments of Collembola and Oribatida were combined in two separate files and re-aligned using default settings. All alignments were gap-free and did not contain any stop-codons. In total, we separately analysed two global datasets that contained all Oribatida and all Collembola, and seven local datasets, one for each genus and one for the family Oppiidae.

Barcoding thresholds were estimated within a range from 1% to 20% distance, at intervals of 1 % for all datasets in R using the threshOpt() function in the *spider v1.5* package (*Brown et al., 2012*). Afterwards, we used the ASAP web application (Assemble Species by Automated Partitioning; https://bioinfo.mnhn.fr/abi/public/asap/; *Puillandre, Brouillet & Achaz, 2021*) to check the potential number of partitions and the size of the corresponding barcoding gap. We provided the sequence alignment, selected the K2P parameter as model of sequence evolution and kept the remaining parameter as default settings. This method is an improved version of the Automated Barcode Gap Discovery (ABGD; *Puillandre et al., 2012*), which partitions single-locus datasets into hypothetical species by re-iteratively finding the best partitions that separate nominal species in the dataset using genetic distances. Different from ABGD, this version does not require *a*

**Table 1  Summary of Oribatida and Collembola used in this study for identifying a barcoding gap in soil-living invertebrates.** Bold taxa have the broadest and densest geographic sampling within the investigated genus and sampling range is comparable among all genera, except for Oppiidae, which covered a smaller sampling area. Accession numbers of specimens are provided in the alignments in the Supplementary Material. The column ASAP refers to the number of genetic lineages (subsets) for each species detected by the ASAP analysis (see Table 4). One or more individuals of species marked with asterisk (*) were assigned to the same genetic lineage (ASAP subset).

| Taxon | Congeneric | No. inds. | ASAP | Taxon | Congeneric | No. inds. | ASAP |
|---|---|---|---|---|---|---|---|
| All Oribatida | | 853 | | All Collembola | | 612 | |
| Achipteria | *A. catskillensis* | 11 | 1 | Ceratophysella | *C. bengtssonii* | 20 | 1 |
| | ***A. coleoptrata*** | **138** | 12 | | *C. communis* | 31 | 2 |
| | *A. howardi* | 4 | 1 | | *C. comosa* | 7 | 1 |
| | Total | 153 | 14 | | ***C. denticulata*** | **60** | 7 |
| Nothrus | *N. anauniensis* | 20 | 1 | | *C. granulata* | 7 | 2 |
| | *N. borussicus* | 8 | 1 | | *C. liguldorsi* | 12 | 2 |
| | *N. palustris* | 10 | 2 | | *C. longispina* | 59 | 1 |
| | *N. pratensis* | 5 | 2 | | *C. pseudarmata* | 44 | 2 |
| | ***N. silvestris*** | **100** | 1 | | *C. scotica* | 4 | 1 |
| | Total | 143 | 7 | | *C. skarzynskii* | 17 | 1 |
| Platynothrus | *P. capillatus* | 4 | 1 | | *C. succinea* | 5 | 1 |
| | ***P. peltifer*** | **160** | 3 | | Total | 266 | 21 |
| | *P. thori* | 4 | 1 | Folsomia | *F. bisetosa* | 15 | *3 |
| | *P. yamasakii* | 81 | 1 | | *F. candida* | 47 | 3 |
| | Total | 249 | 6 | | *F. ciliata* | 6 | 1 |
| Oppiidae | *Aeroppia* sp. | 6 | 1 | | *F. fimentaria (incl. L1-L3)* | 28 | 5 |
| | *Berniniella hauseri* | 2 | 1 | | *F. nivalis* | 39 | 1 |
| | *Dissorhina ornata* | 11 | 2 | | *F. octoculata* | 7 | *3 |
| | *Multioppia* sp. | 6 | 1 | | *F. peniculata* | 15 | 2 |
| | *Oppia nitens* | 92 | *3 | | ***F. quadrioculata*** | **166** | 24 |
| | *Oppia* sp. | 3 | *2 | | *F. sexoculata* | 23 | 2 |
| | ***Oppiella nova*** | **110** | 9 | | Total | 346 | 43 |
| | *Oppiella subpectinata* | 1 | 1 | | | | |
| | *Oppiella uliginosa* | 3 | 1 | | | | |
| | *Ramusella insculpta* | 3 | 1 | | | | |
| | Total | 237 | 24 | | | | |
| Steganacarus | *S. applicatus* | 14 | *2 | | | | |
| | *S. carinatus* | 8 | *2 | | | | |
| | *S. crassisetosus* | 6 | 1 | | | | |
| | ***S. magnus*** | **140** | 18 | | | | |
| | *S. similis* | 5 | 1 | | | | |
| | *S. spinosus* | 15 | 1 | | | | |
| | Total | 188 | 25 | | | | |

*priori* values and provides scores for each partition, which helps users to identify the best partition. Intra- and interspecific genetic distances (corrected with K2P) were plotted with *ggplot2* (*Wickham, 2016*) and *gridExtra* (*Auguie, 2017*) to visualize the barcoding gap. Two plots were generated for all datasets, one using the species names (morphotype) for inter- and intraspecific assignment and one in which species names were replaced by the
number of subsets estimated by ASAP, which equal the number of hypothetical (cryptic) species. The two plots visualize the barcoding gap based on morphological and genetic partitions, respectively. Alternative visualizations for analysing intra- and interspecific genetic distances are histograms (Figs. S1–S2) and scatterplots (*Phillips, Gillis & Hanner, 2022*; Figs. S3–S4) and are provided in the Supplementary Material.

Additionally, we calculated a Maximum Likelihood (ML) tree for the complete Collembola and Oribatida datasets to check if a character-based method accomplishes accurate species assignment in terms of reciprocal monophyly among species. If the distance-based methods (threshold optimization and ASAP) ignore important diagnostic characters in the datasets, this would be a meaningful alternative method (*DeSalle, Egan & Siddal, 2005*). In contrast to Neighbor Joining, which relies on a genetic distance matrix, ML uses each position in a sequence alignment to infer relationships among taxa. For character-based analyses, all datasets were collapsed to haplotypes using FaBox Haplotype Collapser (*Villesen, 2007*) to exclude identical sequences and to reduce the number of sequences to informative taxa for the phylogenetic tree construction. Maximum Likelihood trees with 500 bootstrap replicates were calculated for the global Collembola and global Oribatida and the 28S rDNA datasets (*Oppiella, Ceratophysella, Folsomia*) using the *phangorn v2.11.1* package (*Schliep, 2011*; *Schliep et al., 2017*). This approach is quicker than analyses using GMYC or PTP. However, it only enables to check for monophyly of taxa based on tree topology and statistical support values (bootstraps) of clades. For formal species delineation GMYC or PTP methods are recommended, but this was not our focus. Setting a starting point for the ML optimization requires a genetic distance matrix and a Neighbor Joining tree, which were calculated with dist.dna() and bionj() in R using observed distances (model ="raw"). The ML tree was calculated using the optim.pml() function, as model of sequence evolution we selected the standard model K2P (model ="K80"). The analysis did not include any outgroups and trees remained unrooted.

### Representativeness of haplotype diversity in datasets

We also performed a rarefaction analysis to quantify the representativeness of the sample sizes for species haplotype diversity. Rarefaction was conducted for all haplotypes for which we had geographic sampling information (Collembola: *C. denticulata, F. quadrioculata*; Oribatida: *A. coleoptrata, N. silvestris, O. nova, P. peltifer, S. magnus*). The analysis was performed with the *iNEXT v3.0* package (*Chao et al., 2014a*; *Chao et al., 2014b*; *Hsie, Ma & Chao, 2022*) in R with 1,000 bootstrap replicates, using species richness ($q = 0$) and exponential Shannon entropy ($q = 1$) as measures of diversity.

## RESULTS

Datasets included 970 Oribatida and 612 Collembola sequences of COI, and alignments were between 507 bp and 657 bp long (Table 2), and covered the standard barcoding region of COI. Only a few sequences were below 500 bp long, predominantly in the Oribatida genus *Steganacarus* and the Collembola genus *Folsomia*.

**Table 2  Summary statistics of datasets.** (A) Information on number of genera and species per taxon, the minimum, maximum, median and mean number of sequences used. Oppiidae were analysed on a higher taxonomic level, *i.e.*, at genus instead of species level. (B) Sequence information of datasets, giving the number of sequences, the length of the alignment, the minimum, maximum, median mean and median length of sequences and the number of sequences that were below 500 bp per taxon.

**(A)**

| | | | | Taxa information | | |
|---|---|---|---|---|---|---|
| | Genera | Species | Min | Max | Median | Mean |
| All Oribatida | 10 | 28 | 1 | 160 | 8 | 35 |
| *Achipteria* | 1 | 3 | 4 | 138 | 11 | 51 |
| *Nothrus* | 1 | 5 | 5 | 100 | 10 | 29 |
| *Platynothrus* | 1 | 4 | 4 | 160 | 42 | 62 |
| *Steganacarus* | 1 | 6 | 5 | 140 | 11 | 31 |
| All Collembola | 2 | 20 | 4 | 166 | 18 | 31 |
| *Ceratophysella* | 1 | 11 | 4 | 60 | 17 | 24 |
| *Folsomia* | 1 | 9 | 6 | 166 | 23 | 38 |
| | Families | Genera | | | | |
| Oppiidae | 1 | 7 | 2 | 114 | 6 | 34 |

**(B)**

| | | | Alignment information (bp) | | | | |
|---|---|---|---|---|---|---|---|
| | No. sequences | Length | Min | Max | Mean | Median | No. sequences <500 bp |
| All Oribatida | 970 | 657 | 371 | 657 | 565 | 558 | 32 |
| *Achipteria* | 153 | 507 | 507 | 507 | 507 | 507 | 0 |
| *Nothrus* | 143 | 580 | 371 | 580 | 572 | 580 | 2 |
| *Platynothrus* | 249 | 558 | 417 | 558 | 544 | 558 | 8 |
| Ditto, *Steganacarus* | 188 | 591 | 476 | 591 | 551 | 526 | 21 |
| All Collembola | 612 | 583 | 310 | 583 | 564 | 583 | 48 |
| *Ceratophysella* | 266 | 651 | 485 | 651 | 642 | 651 | 1 |
| *Folsomia* | 346 | 583 | 310 | 583 | 552 | 583 | 47 |
| Oppiidae | 237 | 657 | 459 | 657 | 632 | 657 | 1 |

## Barcoding gap threshold detection for different genetic distances

The global datasets (all Oribatida, all Collembola) had relatively high cumulative errors (false positives and false negatives, Fig. 1; Table 3). The optimized local barcoding threshold for the local datasets differed among taxa (Fig. 1; Table 3). One dataset had a narrow threshold without species mismatches (*Achipteria*, 15%), others had large threshold ranges without mismatches (*Nothrus, Platynothrus* and *Ceratophysella*), and for the remaining datasets it was not possible to define a barcoding threshold without any mismatches (Oppiidae, *Steganacarus, Folsomia*). The optimal barcode thresholds at genus level exceeded the standard barcoding threshold of 2%–3% by 1% (*Platynothrus*) and up to 6% (*Achipteria*), except in the two genera *Nothrus* and *Ceratophysella*.

## Distance-based specimen assignment with ASAP

The ASAP algorithm provides scores for the ten most probable partitions. For all datasets, and in all partitions, ASAP found more subsets than nominal species, *i.e.,* the datasets likely contained more (*i.e.,* cryptic) species than were morphologically determined (Table 4). The ASAP partition with the smallest number of subsets increased the number of morphological

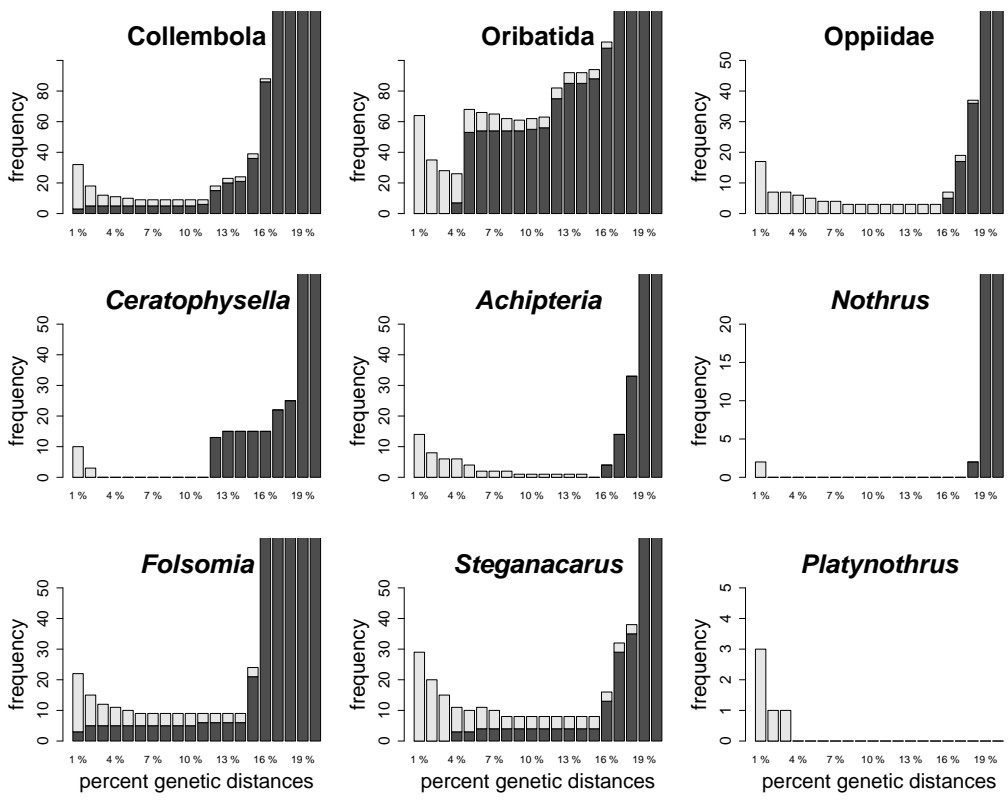

**Figure 1** **Summary of the barcoding threshold optimization of the global and local datasets.** Threshold between 1% and 20% genetic distances were analysed at intervals of 1%. Light grey bars indicate the number of false positive (no conspecific matches within threshold of query), dark grey bars are false negatives (non-conspecific species match within threshold distance of query) of the species assignments for the respective threshold (*x*-axis). Note the different scale of the *y*-axis.

**Table 3** **Range of barcoding gap thresholds and cumulative errors for all datasets.** The cumulative error is the sum of false positives and false negatives. Except for *Nothrus* and *Ceratophysella* the barcoding gap is not present in the investigated species, or exceeds the standard threshold of 2%–3%.

| | Optimal barcoding gap threshold | |
| --- | --- | --- |
| | **Cumulative error = 0** | **Smallest cumulative error** |
| All Oribatida | – | 4% (error: 26) |
| *Achipteria* | 15% | 9%–14% (error: 1) |
| *Steganacarus* | – | 8%–15% (error: 8) |
| *Nothrus* | 2%–17% | 1%, 18% (error: 2) |
| *Platynothrus* | >4% | 2%–3% (error: 1) |
| Oppiidae | – | 8%–15% (error: 3) |
| All Collembola | – | 6%–11% (error: 9) |
| *Ceratophysella* | 3%–11% | 2% (error: 3) |
| *Folsomia* | – | 6%–14% (error: 9) |

**Table 4 Summary of the estimated number of genetic lineages for each local dataset.** The number of morphologically determined species (No. of species) is given for each dataset together with the number of genetic lineages (No. of subsets) estimated by ASAP. For each dataset, the partition with the lowest number of subsets and the highest ranks was selected. The respective scores (including ranks) and statistical support are provided, along with the estimated genetic distance threshold (Threshold distance) that separates the individual subsets. For a detailed list of subsets per species refer to Table 1.

| | No. of species | No. of subsets | ASAP-score | *P*-value (rank) | W (rank) | Threshold distance [%] |
|---|---|---|---|---|---|---|
| All Oribatida | 28 | 69 | 25.0 | 1.20e−04 (4) | 8.12e−06 (46) | 15.0 |
| *Achipteria* | 3 | 14 | 5.5 | 2.99e−02 (3) | 7.13e−04 (8) | 6.9 |
| *Nothrus* | 5 | 7 | 2.0 | 1.00e−05 (2) | 5.53e−04 | 12.2 |
| *Platynothrus* | 4 | 6 | 4.5 | 1.00e−05 (1) | 3.39e−05 (3) | 15.0 |
| *Steganacarus* | 6 | 25 | 3.5 | 1.60e−04 (1) | 1.79e−04 (6) | 15.0 |
| Oppiidae | 10 | 24 | 3.0 | 3.00e−04 (1) | 3.17e−04 (5) | 15.8 |
| All Collembola | 20 | 64 | 12.5 | 1.00e−05 (2) | 6.28e−05 (23) | 11.3 |
| *Ceratophysella* | 11 | 21 | 5.5 | 2.02e−03 (4) | 3.93e−04 (7) | 13.8 |
| *Folsomia* | 9 | 43 | 4.5 | 1.00e−05 (1) | 1.70e−04 (8) | 8.0 |

species to hypothetical species (or genetic lineages) from 5 to 7 (*Nothrus, Platynothrus*), from 11 to 21 (*Ceratophysella*), from 3 to 14 (*Achipteria*), from 6 to 25 (*Steganacarus*) and from 9 to 43 (*Folsomia*). The highest numbers of hypothetical species, or additional genetic lineages, were detected in species with the densest sampling, *i.e., A. coleoptrata, S. magnus, C. denticulata* and *F. quadrioculata* (Table 1). Interestingly, in the two parthenogenetic genera *Nothrus* and *Platynothrus*, only two additional hypothetical species were detected by ASAP, which is little compared to the other genera.

## Distance-based barcoding gap with ASAP

Accurate specimen assignment requires a gap between the largest genetic distance within and the smallest distance between species. We compared the distribution of intra- and interspecific distances of nominal species with that of the genetic lineages inferred by ASAP (Fig. 2), selecting the partitions with the least number of subsets. Our analysis consistently demonstrated that genetic distances of COI within and between morphologically assigned species overlap, which makes accurate species assignment impossible. The parthenogenetic oribatid mite genus *Nothrus* was a single exception, which showed a clear barcoding gap for morphologically assigned species. When genetic lineages (ASAP subsets) were considered, a barcoding gap between intra- and interspecific distances was present. Overall, the assignment of genetic lineages to morphospecies reduced the overlap of intra- and interspecific genetic distances considerably in all datasets, However, the effect was much more pronounced in Collembola than Oribatida and generated a barcoding gap that spanned a range of more than 10% between intra- and interspecific genetic distances. Among Oribatida, the effect of splitting morphotypes into genetic lineages was not strong. However, for two Oribatida datasets, *Achipteria* and *Platynothrus*, the choice of using the partition with the lowest number of subsets was too conservative because the resulting barcoding gap was very narrow. The barcoding gap thresholds estimated for the partition with the lowest number of subsets ranged in Oribatida 6.9% (*Achipteria*), 12.2% (*Nothrus*), 15.0% (*Steganacarus, Platynothrus*, all Oribatida) and 15.8% (Oppiidae); in

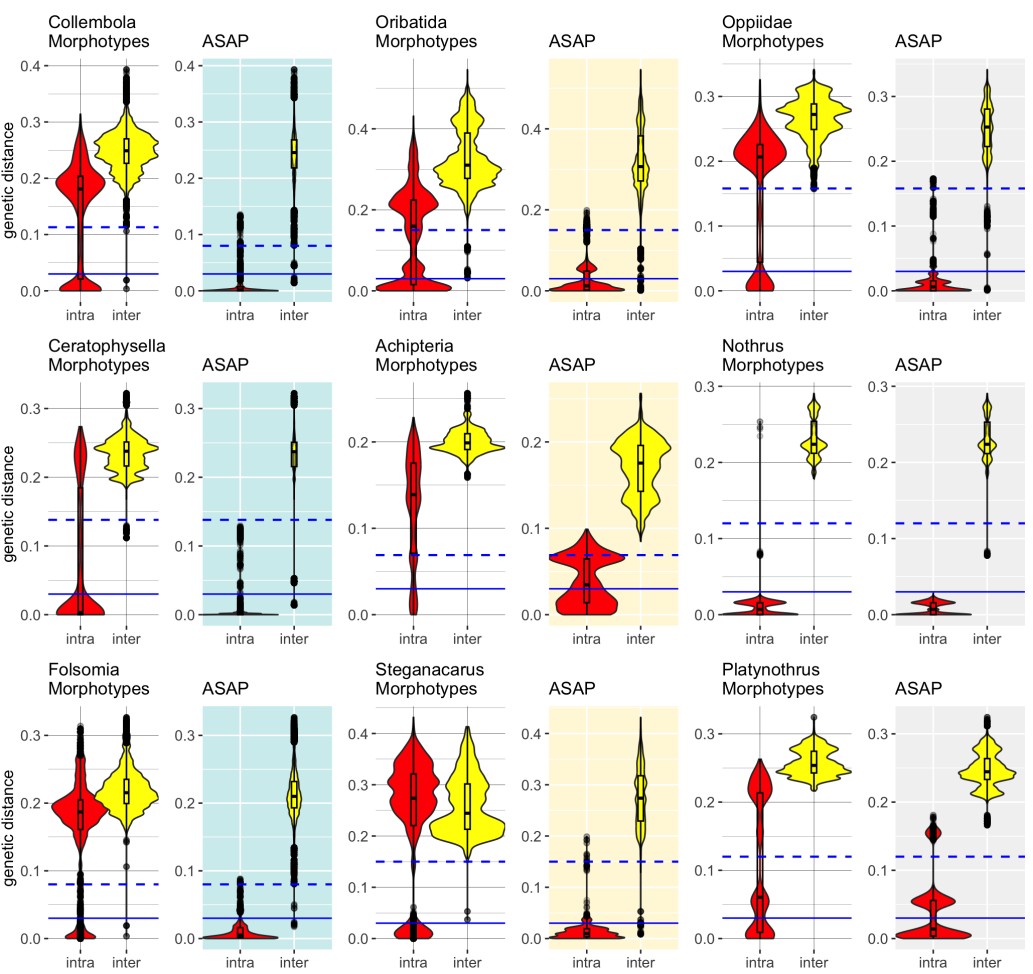

**Figure 2** **Distribution of intra- (red violins) and interspecific (yellow violins) genetic distances in morphological and genetic entities in Collembola and Oribatida.** Distances were calculated for each dataset based on the nominal species names (Morphotypes) and using the same dataset but assigning sequences to genetic lineages (ASAP). The ASAP partition with the smallest number of subsets was used to assign genetic lineages. Specimens that overlap in intra- and interspecific distances cannot be assigned accurately to species based on COI. The splitting of the dataset into genetic lineages created a barcoding gap that improved the accuracy of specimen assignment. Solid blue lines indicate the 3% genetic distances threshold, dashed lines represent the genetic distances of the barcoding gap calculated with ASAP for the respective dataset (Collembola 11.3%, *Ceratophysella* 13.8%, *Folsomia* 8%; Oribatida 15%, *Achipteria* 6.9%, *Steganacarus* 15%, Oppiidae 15.8%, *Nothrus* 12%, *Platynothrus* 15%). Notice the different scale of the *y*-axis.

Collembola 8.0% (*Folsomia*), 11.3% (all Collembola) and 13.8% (*Ceratophysella*). Notably, the intraspecific distances of the genetic lineages of *Platynothrus* show three clusters in distribution frequencies (<3%, at 3%–8%, 14%–17%) that likely represent three genetic lineages in *P. peltifer* (Table 1).

The outliers, *i.e.,* single data points scattered within the range of the barcoding gap, likely belong to sequences that were considerably shorter than the average sequences. Both Collembola datasets were more heterogeneous in sequence lengths than the Oribatida datasets. Only the datasets of *Steganacarus* and Oppiidae also had very short sequences

compared to the median sequence lengths, and both also had outliers after splitting morphotypes into genetic lineages. A few outliers remained for the genus *Nothrus*, which likely belonged to the species *N. palustris* and *N. pratensis*. After splitting both species into two genetic lineages as proposed by ASAP, nearly all outliers disappeared.

## Character-based specimen assignment with Maximum Likelihood

Reliability of specimen assignment based on phylogenetic inference and therefore on molecular characters was very poor in Collembola (Fig. 3). The two genera *Ceratophysella* and *Folsomia* and the species within each genus were not monophyletic. Species clustered within clades of other species several times. The topology of Oribatida supported monophyly for most genera (Fig. 4). The species in the genus *Nothrus* and *Platynothrus* were also monophyletic. Only *Nothrus pratensis* separated into two highly supported, non-monophyletic clades. Within the genus *Steganacarus* all species were monophyletic, except *S. magnus* which formed five clades, two with 100% bootstrap support. The species *S. carinatus* was monophyletic, but separated into two highly supported clades. One sequence of *S. applicatus* clustered within *S. carinatus* while the remaining sequences were monophyletic with very high support. Possibly this single sequence represents a misidentified individual. Most genera of Oppiidae were monophyletic, except for *Disorrhina* and *Oppia*. The sequences assigned to *Oppia* sp. were sister to one clade of *O. nitens* with very high bootstrap support. It is possible that these sequences belong to the species *O. nitens*. All remaining species were monopohyletic with 100% bootstrap support. The genus *Achipteria* was represented by only three species. The two species *A. howardi* and *A. catskillensis* were monophyletic, but the sequences of *A. coleoptrata* were non-monophyletic.

## Nuclear gene

The uncorrected p-distances among 28S rDNA in *C. denticulata* were high across all sequences the maximum genetic distances were 5.6%, but the mean distances were only 0.16% (median 3.2%). The different haplotypes corresponded very well with the seven genetic lineages suggested by ASAP (Table S2), *i.e.,* each 28S rDNA haplotype included a single ASAP lineage. However, the two datasets were not entirely congruent, *i.e.,* seven specimens of the 28S rDNA dataset were not represented as COI sequences, and four specimens in the COI dataset were not present in the 28S rDNA dataset. In *F. quadrioculata*, the 24 genetic lineages did not reflect at all the 28S rDNA sequences. The nuclear gene represented only three haplotypes with uncorrected p-distances below 1% (maximum 0.35%, mean 0.16%, median 0.18%). These 28S rDNA haplotypes comprised nine, three and one COI lineages that were identified by ASAP, respectively. Notably, only 56 specimens of 28S rDNA were represented from the 166 specimens of the COI nucleotide dataset. Among *O. nova* p-distances of 28S rDNA were also small, below 2% (maximum 1.98%, mean 0.43%, median 0.29%). In contrast to the two species above, each of the nine genetic lineages of *O. nova* supported by ASAP carried different 28S rDNA haplotypes, *e.g.,* in one common COI lineage that comprised 30 specimens (lineage_1; Table S2), individuals represented twelve (slightly) different 28S rDNA haplotypes.

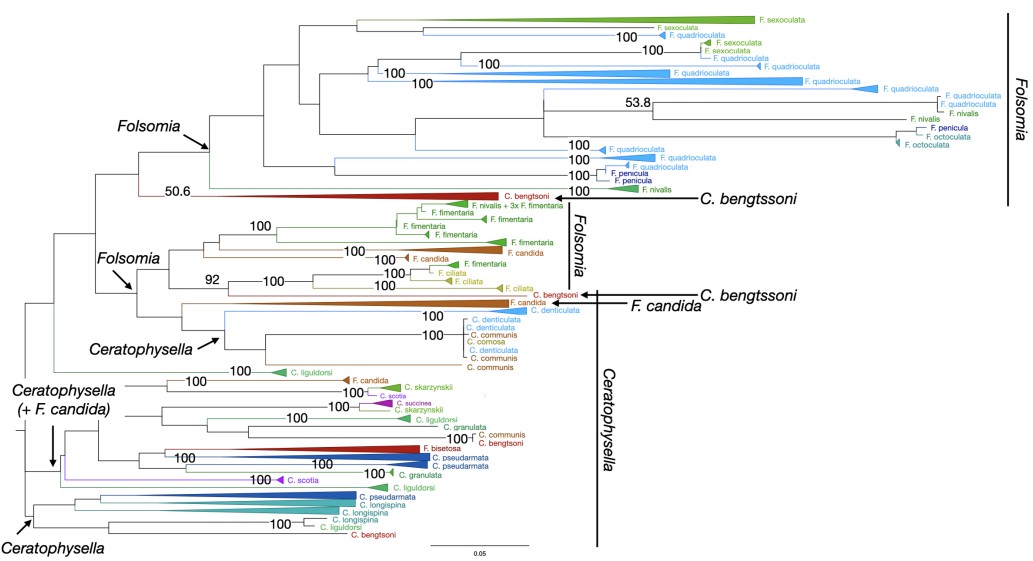

**Figure 3** **Phylogenetic tree of all Collembola for character-based species assignment.** Likelihood tree based on 313 haplotypes of 612 COI sequences and 500 bootstrap replicates. Monophyletic nodes were collapsed, bootstrap values >50% are shown on nodes. The two genera *Ceratophysella* and *Folsomia* are not monophyletic, and species within genera are also not monophyletic.

## Representativeness of sampling effort

Rarefaction curves (Fig. 5) showed that Oribatida species had more haplotypes than Collembola and that sexual Oribatida species (*A. coleoptrata, S. magnus*) had more haplotypes than parthenogenetic Oribatida (*P. peltifer, N. silvestris*). Further, Collembola reached saturation in species diversity at a sampling size of less than 200 individuals (for COI and 28S rDNA), the pattern was similar for the parthenogenetic Oribatida *N. silvestris*. However, the parthenogenetic Oribatida *P. peltifer* and both sexual Oribatida species did not reach saturation at a sampling size of more than 600 individuals and the expected diversity exceeded 200 haplotypes.

## DISCUSSION

This study tested the validity of a barcoding gap and the applicability of the standard barcoding gene COI for species assignment in two of the most species rich and abundant taxa of soil-living invertebrates, Collembola and Oribatida. The analysed datasets comprised two genera of Collembola with eleven and nine species, respectively. Oribatida datasets comprised four genera with three to six species per genus, and one family-level dataset with ten species in six genera.

Our results showed that correct species assignment was possible within some genera, but not all. However, both distance- and character-based methods were not able to assign species without mismatches when all Collembola or all Oribatida were analysed together. This is likely due to the different ranges of intraspecific genetic distances, demonstrating the absence of a general (global) barcoding gap for COI in these taxa. The genetic divergence separating intra- and interspecific distances differed among taxa and exceeded the standard

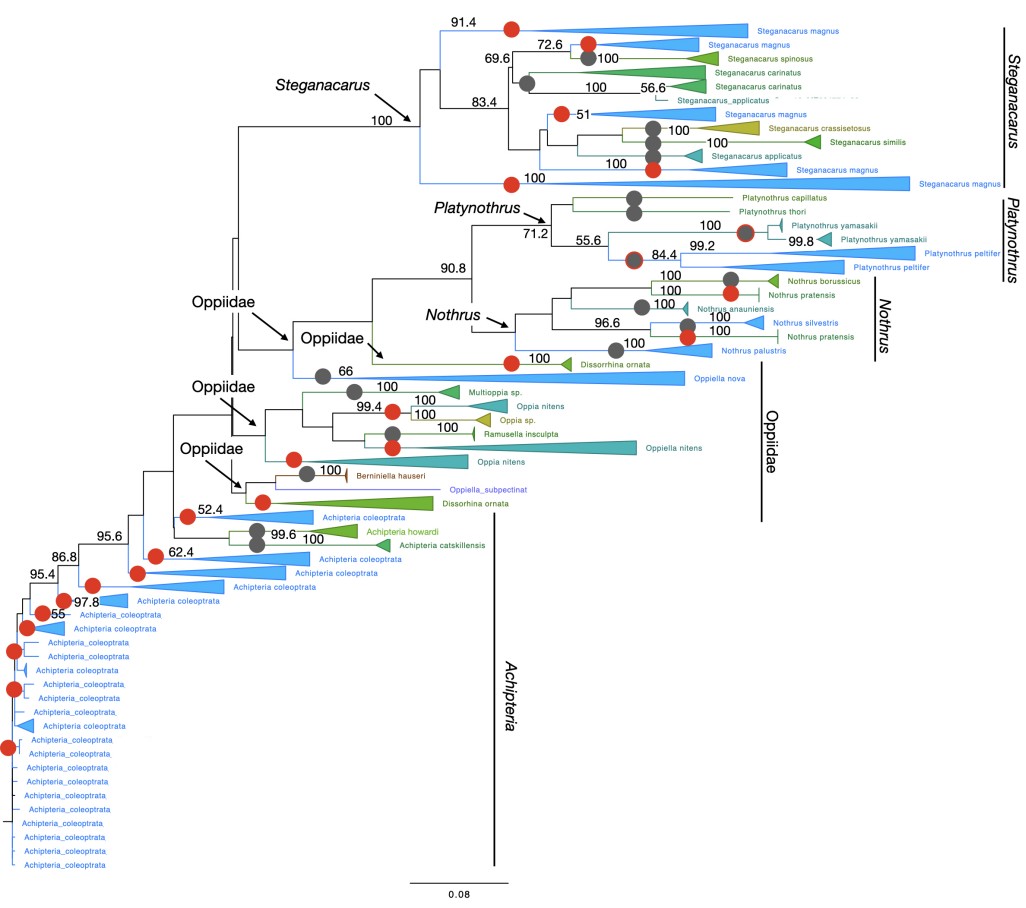

**Figure 4** **Phylogenetic tree of all Oribatida for character-based species assignment.** Likelihood tree based on 514 haplotypes of 970 COI sequences and 500 bootstrap replicates. Monophyletic nodes were collapsed, bootstrap values >50% are shown on nodes. Grey circles on branches highlight monophyletic lineages. Red circles highlight non-monophyletic lineages, indicating species for which character-based species assignment is problematic or equivocal. Grey circles with red outlines indicate species that are monophyletic but split into at least two clades. All genera but *Achipteria, Dissorhina, Oppia* and *Oppiella* are monophyletic. The single sequence of *Oppiella subpectinata* was sister to *Berniniella* and potentially represents a misidentified individual. A phylogenetic tree of 28S rDNA haplotypes is provided in Fig. S5.

species threshold of 3% intraspecific genetic distance in all but one species, indicating that taxon-specific thresholds should be applied for correct specimen assignment (*Phillips, Gillis & Hanner, 2022*). Here, the application of algorithms that dynamically adjust thresholds for sequence clusters, and therefore apply flexible thresholds, could improve species assignment in soil invertebrates (*James, Luczak & Girgis, 2018*; *Chiu & Ong, 2022*).

Absence of a global barcoding gap in the COI gene seems to be particularly relevant for soil-living animals and hampers the application of automated specimen assignments in DNA-based biodiversity surveys such as metabarcoding. Absence of a global barcoding gap had also been demonstrated for Annelida, among which earthworm taxa accounted for one third of interspecific comparisons with 0% genetic divergence (*Kvist, 2016*). In metabarcoding studies, Collembola had a high failure rate and high numbers of false

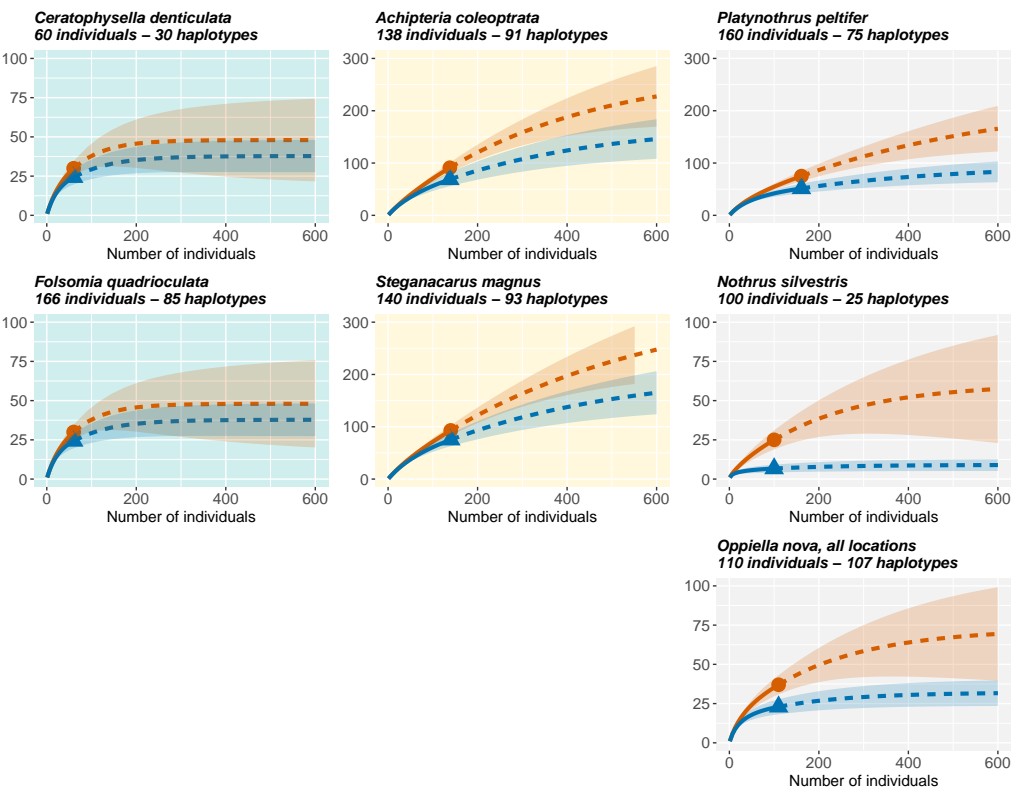

**Figure 5  Rarefaction of Collembola and Oribatida species.** Only species with sampling site information are included for quantifying the representatives of genetic diversity in the different datasets. Collembola species reach soon saturation in haplotype diversity, while sexual Oribatida species (*A. coleoptrata, S. magnus*) do not reach saturation. The parthenogenetic Oribatida species *P. peltifer* also reaches saturation close to a sampling size of 600 individuals, but expected diversity is lower with less than 200 haplotypes (note the different scale of the *y*-axis). The parthenogenetic Oribatida species *N. silvestris* shows the lowest diversity and reaches soon saturation, indicating that sampling size was almost representative for the expected haplotype diversity in this species. Solid lines indicate the rarefaction, dotted lines the extrapolation. The tested diversity measures using *iNEXT* were species richness ($q = 0$, red lines) and Shannon diversity ($q = 1$, blue lines). Notably, the two indices are more similar in Collembola than in Oribatida. Rarefaction plots of 28S rDNA haplotypes are provided in Fig. S6.

positives for species assignments based on public databases and COI (*Recuero, Etzler & Caterino, 2023*). Among mites, specimen assignment is in general correct at least to family and order level (*Oliverio et al., 2018*; *Ustinova et al., 2021*; *Young, deWaard & Hebert, 2021*; *Young & Hebert, 2022*; *Recuero, Etzler & Caterino, 2023*). General explanations for failures in species assignments include the lack of completeness and misidentified individuals in reference databases, geographic underrepresentation of species and a neglect of assigning genetic lineage identities to sequences in reference databases (*Kvist, 2016*; *Martinsson, Rhoden & Erseus, 2016*; *Young et al., 2019*; *Young, deWaard & Hebert, 2021*; *Phillips, Gillis & Hanner, 2022*; *Recuero, Etzler & Caterino, 2023*). The rarefaction analysis demonstrated that genetic diversity is exceptionally high within morphospecies of soil-living invertebrates, and more genetic diversity is to be expected in additional samples. In particular, rarefaction

curves for Oribatida did not reach saturation without sampling hundreds of additional individuals. For Collembola the number of expected COI haplotypes is lower as curves reached saturation at an expected sampling size of about 200 individuals, indicating that required sampling effort can be reached sooner than in Oribatida.

Results of this study provide an additional explanation why molecular species assignment often fails in Collembola and Oribatida. The more detailed analysis of the individual datasets at genus level showed that intra- and interspecific distances of taxa greatly overlapped, demonstrating the absence of a barcoding gap between species for all taxa, except for the parthenogenetic Oribatida genus *Nothrus*. The automated partitioning of datasets based on genetic distances (ASAP) suggested that each morphospecies (except most species within *Nothrus* and *Platynothrus*) consists of several genetic lineages, indicating the presence of putative or cryptic species. After assigning individuals according to genetic lineages, a barcoding gap between intra- and interspecific distances became apparent, but it still exceeded the standard threshold of 3%. Alternative partitions in the species assignment analyses also opted for smaller thresholds, but resulted in even more genetic lineages. From a conservative approach, the two sexual Oribatida species *A. coleoptrata* and *S. magnus* comprised 12 and 18 genetic lineages, respectively, with the relatively high barcoding threshold estimates of 6.9% (*A. coleoptrata*) and 15.0% (*S. magnus*). The Collembola species *C. denticulata* consisted of seven genetic lineages (barcoding gap of 13.8%) and *F. quadrioculata* of 24 genetic lineages (barcoding threshold of 8.0%). Notably, the parthenogenetic Oribatida species *O. nova* separated only into nine genetic lineages, *P. peltifer* into three and *N. silvestris* remained a single species which was consistent with morphological assignments.

In contrast to our hypothesis, the detection of a barcoding gap and thus species delimitation worked well for the parthenogenetic, but not for the sexual taxa. Species boundaries of *Nothrus* were clear and unequivocal. However, intra- and interspecific distances among *Platynothrus* overlapped, likely due the presence of three genetic lineages in *P. peltifer*. This is consistent with previous studies that identified seven genetic lineages in *P. peltifer* based on a transcontinental sampling, and demonstrated that lineages are consistent with species based on the 4x rule of parthenogenetic speciation (*Heethoff et al., 2007*; *Birky & Barraclough, 2009*; *Birky et al., 2010*).

Detection of deeply divergent genetic lineages in morphological consistent species is a common phenomenon and detection rate of cryptic species accelerated with the application of molecular identification tools (*Bickford et al., 2007*; *Pfenninger & Schwenk, 2007*; *Skoracka et al., 2015*; *Struck et al., 2018*). However, it remains important to consider these putative species carefully based on barcoding approaches, as delimitation is based only on a single genetic marker. The putative genetic lineages were highly congruent with nuclear haplotype diversity in *C. denticulata*, but not in *F. quadrioculata* and *O. nova*. Interestingly, genetic variance of nuclear and mitochondrial genes was opposite in the two latter species. In *F. quadrioculata* a single nuclear haplotype comprised many COI lineages, but in *O. nova* a single COI lineage comprised several nuclear haplotypes. This suggests that different selective forces might act on mitochondrial and nuclear genes in the two species. The higher mutation rate of mitochondrial compared to nuclear genes explains

the higher diversity in COI in *F. quadrioculata*, indicating relatively recent divergence of lineages that had not yet been accompanied by variation in the nuclear gene. By contrast, in *O. nova* the mitochondrial gene shows relatively little variation, which likely is related to stronger purifying selection in parthenogenetic Oribatida (*Brandt et al., 2017*; *Brandt et al., 2021*). The two other parthenogenetic species (*N. silvestris* and *P. peltifer*) also show very little genetic variation, unfortunately, no additional genes were available for these taxa.

Our results demonstrate, that soil-living microarthropods comprise deeply divergent genetic lineages. Barcoding or metabarcoding studies based on single genes will therefore likely result in high numbers of unassigned reads or overestimate species numbers and consequently misrepresent species richness in communities. Potential species status should therefore be corroborated with an integrative taxonomic approach using multiple genetic markers and, if possible, re-examination of morphotypes (*Schäffer, Kerschbaumer & Koblmüller, 2019*; *Lienhard & Krisper, 2021*). However, morphological differences are often subtle, making traditional determination of soil microarthropods even more challenging. The nuclear 28S rDNA gene has been proposed as secondary barcoding marker for Oribatida (*Lehmitz & Decker, 2017*), but its applicability in a wider geographic range and different habitats has not been tested. Alternatively, metagenomic studies provide multiple genes per specimen which likely improves accuracy in specimen assignment. However, similar to metabarcoding based on single genes such as COI and/or 28S rDNA, successful application of metagenomics depends on representative reference databases. Notably, single reference genomes, or one/few barcodes per species will not cover intraspecific variation. Species with high intraspecific genetic variance would require "pan-barcodes" *i.e.,* multiple barcodes from individuals that were sequenced across the range of a species to cover the extent of its intraspecific genetic variance.

The limited taxon sampling in this study demonstrates that even for the relatively intensive sequenced COI gene, databases do not provide taxonomic breadth for reliable species delimitation of Collembola and Oribatida. It is possible that species assignment will improve with a better reference database, but it is also important to understand the mechanisms that explain the barcoding gap, *i.e.,* the substantial genetic divergence of COI sequences between closely related Collembola and Oribatida taxa. It is unknown if genetic variance is neutral or adaptive, or if mitonuclear or environmental interactions (*Hill, 2020*) generate the genetic structure in soil-living microarthropods. Fixation of neutral variance is one likely mechanism in the investigated taxa. The high numbers of haplotypes and nucleotide diversity suggest that COI is already highly saturated in these species. Many Collembola and Oribatida species are very abundant in local communities, suggesting high effective population sizes. This could enable the maintenance of neutral allelic variation and blur a barcoding gap in order to maintain the highly conserved protein sequence of COI. Repeated episodes of extreme population bottlenecks can also generate a barcoding gap between species. However, this is unlikely because high genetic variance in general argues against repeated population bottlenecks. However, the Oribatida species *N. silvestris* shows exceptionally low genetic variance compared to the other taxa, and consists of a single genetic lineage. It is possible that the low genetic variance resulted from a bottleneck this species experienced during Quaternary glaciations (<2.6 mya). Molecular divergence times

among genetic lineages in the other species are several million years old, most date back to the Miocene (23-5 mya) and support the accumulation of neutral variance by genetic drift in Oribatida and founder events in Collembola (*Rosenberger et al., 2013*; *von Saltzwedel, Scheu & Schaefer, 2016*). Directional selection on mitochondrial genotypes and disrupted gene flow can lead to rapid divergence among populations. Collembola and in particular Oribatida are poor active dispersers due to their small body size, which reduces gene flow among populations and is a possible explanation for mitochondrial lineages corresponding with nuclear 28S rDNA haplotypes and sampling locations in *C. denticulata* (*Porco et al., 2012b*; *von Saltzwedel, Scheu & Schaefer, 2016*). However, reduced gene flow seems unlikely in *F. quadrioculata* due to the low genetic variance in the nuclear 28S rDNA gene compared to the highly variable mitochondrial COI gene. Genetic distances among lineages suggest maintenance of relatively ancient divergences, which argues against rapid divergence and disrupted gene flow. Further, this explanation does not apply for parthenogenetic species. Apparently, different mechanisms seem to account for the genetic variance in COI within species of Collembola and Oribatida. This is not surprising considering that the species in this study likely are separated by tens to hundreds of millions of years, each having its own evolutionary trajectory (*Schaefer et al., 2010*; *Schaefer & Caruso, 2019*; *Leo et al., 2019*; *Katz, 2020*; *van Straalen, 2021*).

This study showed that metabarcoding using the standard gene COI is problematic when investigating biodiversity of soil invertebrates. Advances in second- and third-generation sequencing technologies can significantly contribute to improve the reliability of barcodes for genetically diverse and potentially cryptic species. Proposed as an alternative to small barcoding fragments, low coverage shotgun sequencing and genome skimming offer increased species discrimination by covering entire organellar genomes and ribosomal sequences (*Coissac et al., 2016*). PacBio sequencing technology generates reads of approximately 3 kb with very low error rates. This enables sequencing of nearly full-length marker genes and their flanking regions, which improves taxonomic resolution and reduces spurious Operational Taxonomic Units (OTUs) (*Tedersoo & Anslan, 2019*). Notably, genomes of Collembola and Oribatida typically range between 350 and 500 Mb, enabling to obtain reasonable sequencing read depth at moderate prices. Further, wet-lab protocols for genome sequencing of small, non-model invertebrates have been developed (*Collins et al., 2023*) and the results underscore the importance of taking intragenomic variance into account in order to integrate genetic and morphological species boundaries. We propose that characterizing pan-genomes is crucial for identifying species in soil invertebrates (*Tettelin et al., 2005*). This approach will also contribute to develop informative barcoding genes (pan-barcodes) in soil invertebrates that lack a distinct barcoding gap. A pan-genome includes the complete set of genes shared by all individuals within a species and consists of conserved (core) and variable (accessory) gene regions (*Golicz et al., 2020*). The core genome covers all genes that are present in all individuals and the accessory genome includes the genomic regions that are variable among species. This variance is often due to ecological, geographical or reproductive boundaries (*Reno et al., 2009*). Accordingly, pan-genomes offer a holistic view of a species' genome, allowing to identify both conserved

and variable regions that are suitable for designing robust barcoding markers, in particular in taxonomically challenging organisms.

## CONCLUSIONS

This study demonstrated that intra- and interspecific genetic divergences in the standard barcoding gene COI overlap in several species of Collembola and Oribatida. This is violating the assumption of a barcoding gap, which is a precondition for molecular species assignment and questions the applicability of the standard barcoding gene COI for soil-living microarthropods. Further, the presence of deeply divergent genetic lineages within morphologically consistent species emphasizes that (meta-)barcoding results solely based on a single genetic marker should be interpreted carefully. Based on COI, morphologically consistent species comprised numerous cryptic species. Without additional genetic and morphological data, the taxonomic status of these cryptic species is questionable. The assignment of genetic lineages to sequences in reference databases and application of flexible or species-specific thresholds could improve specimen assignment. However, the strong discrepancy between morphological conservativeness and genetic variance of many soil invertebrates calls for a more general approach. We are promoting to develop barcoding approaches with alternative sequencing technologies that generate more genetic data than metabarcoding, such as low-coverage shotgun sequencing of genomes (*e.g.*, genome skimming and metagenomics) or long-read sequencing of marker genes using third generation sequencing technologies. Further, we advocate for the construction and analysis of pan-genomes to understand genetic species boundaries and to develop reliable barcoding markers that cover the whole range of genomic variance of species (pan-barcodes). Regardless of the approach taken, it is essential for reference databases to cover the intraspecific variability of a species throughout its geographic range.

### Funding

Jéhan Le Cadre was funded by the IMABEE master-programme at University of Göttingen. Ina Schaefer and Miklós Bálint are funded by the Landes-Offensive zur Entwicklung Wissenschaftlich-ökonomischer Exzellenz (LOEWE) Program of the Hessian Ministry of Higher Education, Research, Science and the Arts through the LOEWE Centre for Translational Biodiversity Genomics (LOEWE-TBG). Financial support was provided by the German Science Foundation (DFG)(SCHE 376/38-3). We acknowledge support by the Open Access Publication Funds of the University of Göttingen. The funders had no role in study design, data collection and analysis, decision to publish, or preparation of the manuscript.

### Grant Disclosures

The following grant information was disclosed by the authors:
The IMABEE master-programme at University of Göttingen.

The Landes-Offensive zur Entwicklung Wissenschaftlich-ökonomischer Exzellenz (LOEWE) Program of the Hessian Ministry of Higher Education, Research, Science and the Arts through the LOEWE Centre for Translational Biodiversity Genomics (LOEWE-TBG). The German Science Foundation (DFG)(SCHE 376/38-3).
The University of Göttingen.

## Competing Interests

The authors declare there are no competing interests.

## Author Contributions

- Jéhan Le Cadre conceived and designed the experiments, performed the experiments, analyzed the data, prepared figures and/or tables, authored or reviewed drafts of the article, and approved the final draft.
- Finn Luca Klemp performed the experiments, analyzed the data, prepared figures and/or tables, and approved the final draft.
- Miklós Bálint conceived and designed the experiments, authored or reviewed drafts of the article, and approved the final draft.
- Stefan Scheu conceived and designed the experiments, authored or reviewed drafts of the article, and approved the final draft.
- Ina Schaefer conceived and designed the experiments, analyzed the data, prepared figures and/or tables, authored or reviewed drafts of the article, and approved the final draft.

## Data Availability

All alignments and the R code (Rmarkdown) used for analysis and graphics are available at FigShare: Schaefer, Ina; Le Cadre, Jéhan (2024). LeCadre et al PeerJ_R.zip. figshare. Dataset. https://doi.org/10.6084/m9.figshare.25260190.v1.

## Supplemental Information

Supplemental information for this article can be found online at http://dx.doi.org/10.7717/peerj.17709#supplemental-information.

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
