# Peer review of "Applicability and perspectives for DNA barcoding of soil invertebrates"

_PeerJ, doi:10.7717/peerj.17709_

## Round 0.1 · original submission · Minor Revisions

Overall an excellent manuscript. Some minor revisions are necessary. Both reviewers highlight that additional description is needed of the phylogenetic methods used, including model selection. both reviewers note some typographical errors. Please perform a thorough spelling and grammar check before finalising the manuscript.

**Language Note:** The Academic Editor has identified that the English language must be improved. PeerJ can provide language editing services - please contact us at copyediting@peerj.com for pricing (be sure to provide your manuscript number and title). Alternatively, you should make your own arrangements to improve the language quality and provide details in your response letter. – PeerJ Staff

·

Basic reporting

This was a clear and well-written manuscript. I have made a few small grammatical corrections or suggestions throughout, but of a very minor nature. The background was quite thorough - the one area I would like to see some additional details in the introduction is on the concepts and pros/cons of various species-delimitation methods. There was a strong introduction to the concept of a barcoding gap, but very little introduction to other approaches. As well, even though they were not included in the study, it would be informative to give an overview of other commonly used methods such as the BOLD bin assignation, or GMYC. As well, as noted in the attached manuscript, several sentences in the introduction should have been included in Methods instead.

Experimental design

This is an important paper. As metabarcoding studies for soil mesofauna become increasingly common, there is a definite trend of researchers accepting species designations at face value. Studies such as this that critically evaluate the underlying assumptions are greatly needed - not to discount the results of metabarcoding work, but to ensure there is a strong foundation of the caveats and issues that can arise in these types of studies. The methods are clearly described. The one area I would like to see more detail is in the choice of phylogenetic method and the tree-building decisions (e.g. model-selection).

Validity of the findings

The conclusions drawn from the data seem quite solid. The writing of the discussion could perhaps be tightened up a bit, but it is generally thorough and clear. The one area that could have been addressed more explicitly is the validity of the morphological identifications. While the idea of cryptic species was touched upon, in general the results and discussion assumed that the morphologically identified taxa were "correct". I would have liked to see a bit more nuance, wherein the lack of clear species designation based on automated methods may be a "failure" of DNA or of the analytical methods, but alternatively could be a call to do more detailed morphological analysis or integrated taxonomy. Or alternatively, if the authors truly had no doubt in the identity of the named species in this study, they should have justified this confidence.

Additional comments

Overall, this is an excellent and highly relevant paper, and I look forward to its eventual publication.

Reviewer 2 ·

Basic reporting

The authors present a very interesting and timely study on soil invertebrate DNA-barcoding. Although soil invertebrates represent one of the most species-rich taxonomic groups, they remain understudied so far. While barcoding records are slowly accumulating, barcoding gap thresholds may need to be established so that automated species identification (i.e., DNA-metabarcoding) will be easier, more reliable, and more informative in the future. This manuscript explores exactly this topic by using springtail and mite DNA-barcodes to test whether a general barcoding gap for species identification exists or if these need to be established for different taxonomic groups. Results indicate that COI alone will likely be insufficient to reliable species identification and that at the minimum, additional gene regions should be used to confirm species identification in soil invertebrates, particularly, springtails and mites, to improve the future applicability of barcode reference databases for DNA-metabarcoding.

I enjoyed reading this manuscript! It is well-written with clearly formulated goals/research questions that are backed by the authors’ analyses. Literature references are all relevant and the background section explains clearly the rationale of the paper. Article structure and figures are great, too.

Experimental design

The manuscript presents different analytical approaches and explores the level of overlap between intra-and interspecific lineages and thereby, demonstrates that depending on methods and thresholds used, the number of potential taxonomic groups is a) not always well represented by COI alone, and b) that traditional DNA-barcoding thresholds may be not generally applicable to these taxonomic groups. In addition, analyses suggest cryptic biodiversity in the studied taxa, which is relevant for further biodiversity studies and that haplotype diversity was not covered for certain species, suggesting that more sampling is clearly required for most soil invertebrate groups. .

The methods are largely described in sufficient detail (but see below) and several analyses back up the presented conclusions.

The one thing that I would like the authors to consider is to potentially provide more information on the phylogenetic analysis. The authors provide the name of the function they used but were there other parameters that were defined, for example?

Validity of the findings

Conclusions and recommendations are clearly formulated and backed by the findings.

I'm not sure whether all underlying data has been provided. If this includes used DNA sequence data, then probably not. However, I'm unsure if this is required in this case and leave it to the editor to decide.

Additional comments

Line 135, page 11: should ‘Muller’s ratcher’ be ‘Muller’s ratchet’?
Line 229, page 14: Could you please add a version number for the spider package, if available?
Line 236, page 14: Change ‘inter- and interspecific’ to inter- and intraspecific’.
Line 242, page 15: Change ‘Supplementary’ to ‘Supplementary Material’.
Line 248, page 15: Could you please add a version number for the phangorn package, if available?
Line 252, page 15: Could you please add a version number for the iNEXT package, if available?
Line 323, page 17: Change ‘One sequences’ to ‘One sequence’.
Line 529, page 24: Missing word? Change ‘we advocate the construction’ to ‘we advocate for the construction’.
Line 536, page 25: Change ‘Enumarating soil biodiversity’ to ‘Enumerating soil biodiversity’.
Line 573, page 25: Change ‘ancnient’ to ‘ancient’.
Line 576, page 25: Change ‘identify’ to ‘identity’.
Lines 799-801, page 30: I believe the paper title is missing here. Please check and revise.
Line 844, Page 31: Change ‘Unerathing’ to ‘Unearthing’.

---

## Round 0.2 · accepted · Accept

Authors have adequately addressed the major concerns raised in reviews.

·

Basic reporting

This was a clear and well-written manuscript. I appreciate the changes the authors made to the previous version and have no concerns with the structure or content.

Experimental design

This is an important paper. As metabarcoding studies for soil mesofauna become increasingly common, there is a definite trend of researchers accepting species designations at face value. Studies such as this that critically evaluate the underlying assumptions are greatly needed - not to discount the results of metabarcoding work, but to ensure there is a strong foundation of the caveats and issues that can arise in these types of studies. The methods are clearly described, and I appreciate the additional details provided regarding the phylogenetic analysis.

Validity of the findings

While I still feel there could be a more nuanced view of the validity of morphological identification (in particular discussion of cryptic species), it is not something I need to quibble with. I appreciate that the taxa used in this study are well-known and were identified by experts in the field.

Additional comments

I appreciate the care and attention the authors made to the previous round of revisions. This is an important and relevant paper for the quickly expanding field for mesofaunal metabarcoding, and I look forward to seeing it published in PeerJ.